# Analysis of Peri-Implantitis Photothermal Therapy Effect According to Laser Irradiation Location and Angle: A Numerical Approach

**DOI:** 10.3390/biomedicines12091976

**Published:** 2024-09-01

**Authors:** Donghyuk Kim, Hyunjung Kim, Hee-Sun Kim

**Affiliations:** 1Department of Mechanical Engineering, Ajou University, Suwon-si 16499, Republic of Korea; kimdonghyuk20@ajou.ac.kr; 2Department of Dentistry, SMG SNU Boramae Medical Center, Seoul 07061, Republic of Korea

**Keywords:** heat transfer, peri-implantitis, Arrhenius damage integral, photothermal therapy, thermal damage, Arrhenius thermal damage ratio

## Abstract

In recent years, dental implants have become increasingly popular around the world. However, if the implant is not properly managed, inflammation may occur, and the implant itself may need to be removed. Peri-implantitis is a common inflammation that occurs in dental implants, and various laser treatments have recently been studied to eliminate it. In this study, the situation of removing peri-implantitis using photothermal therapy, one of the various laser treatments, was analyzed theoretically and numerically. The temperature distribution in the tissue for various laser irradiation locations, angles, and power was calculated based on heat transfer theory, and the degree of thermal damage to tissue was analyzed using the Arrhenius damage integral. In addition, the thermally damaged region ratio of inflamed and normal tissue was analyzed using the Arrhenius thermal damage ratio and normal tissue Arrhenius thermal damage ratio to confirm the trend of treatment results for each treatment condition. The results of the study showed that if only the thermal damage to the inflamed tissue is considered, the laser should be angled vertically, and the laser should be applied to the center of the inflamed tissue rather than close to the implant. However, if the thermal damage to the surrounding normal tissue is also considered, it was found that the laser should be applied at 1.0 mm from the right end of the inflamed tissue for maximum effect. This will allow for more accurate clinical treatment of peri-implantitis in the future.

## 1. Introduction

With the advancement of medical technology, methods to compensate for tooth loss are increasing, one of which is implant placement [1,2,3]. However, even though dental implants can compensate for tooth loss, improper maintenance can lead to inflammation, which may necessitate the removal of the implants themselves [4,5]. The most common inflammation is peri-implantitis, which is inflammation that occurs around the implant due to food particles, alcohol, smoking, etc., [6]. If peri-implantitis is left untreated for a long period of time, it causes damage to the surrounding gum tissue and alveolar bone [7]. If the damage is excessive and the implant must be removed, reimplantation takes a long time and costs a lot of money. Accordingly, prevention and treatment of peri-implantitis are very important [8,9].

Methods for removing peri-implantitis include physical removal and chemical removal [10,11]. However, physical removal has the disadvantage that it can be difficult to remove spots that are difficult to see, and chemical removal can have side effects [12,13]. To compensate for these drawbacks, removal methods using dental lasers are currently being studied in the dental field [14]. Among the various laser treatments, photothermal therapy (PTT) is a treatment method based on the photothermal effect, which has the advantages of no bleeding and fast recovery [15,16].

Various studies are being conducted on the removal of peri-implantitis using lasers. Hu et al. [17] analyzed the treatment of peri-implantitis by combining conventional non-surgical treatments with various lasers. The results were analyzed using probing depth (PD), plaque index (PLI), clinical attachment level (CAL), and sulcus bleeding index (SBI). The comparison showed that the diode laser was effective in reducing PD, and the Er:YAG laser was effective in terms of PLI, CAL, and SBI. Chen et al. [18] compared the effectiveness of mechanical debridement with the treatment of peri-implantitis using an Er:YAG laser. Results were analyzed for a total of 23 patients, and PD, bleeding on probing (BOP), marginal bone loss (MBL), and anaerobic bacteria counts were confirmed at each elapsed time after treatment. The analysis confirmed a significant reduction in PD in both groups, with the treatment modality utilizing the Er:YAG laser leading to a higher PD reduction compared to mechanical debridement.

The field of photothermal therapy is still mainly focused on tumor removal. Kim et al. [19] confirmed the temperature distribution in tissues according to the number of injections and radius of gold nanoparticles, which are one of the photothermal agents. The temperature distribution in tissues was calculated based on the heat transfer theory, and the results were analyzed focusing on the temperature range where apoptosis occurred. The results showed that the best effect was achieved when the gold nanoparticles were injected seven times, and the laser power was 52 mW. Cheong et al. [20] studied the effects of photothermal agent distribution by performing photothermal therapy on tumors that occurred in the bladder. The temperature was calculated numerically for various distributions of photothermal agents in the tumor. When photothermal agents were located in the tumor center, the temperature rose only in the area in the tumor center. However, when photothermal agents were injected to surround the tumor and go underneath the tumor, it was found that under certain laser irradiation conditions, all tumors were removed.

Most general peri-implantitis laser treatment studies simply analyze treatment results, and theoretical analysis based on heat transfer is insufficient. Additionally, there is a lack of analysis of the temperature rise and distribution of peri-implantitis, which can vary in different treatment situations. Therefore, this study analyzed the temperature distribution of inflamed and surrounding tissues under different laser irradiation locations, angles, and power for peri-implantitis photothermal therapy theoretically and numerically. The results were then applied to the Arrhenius damage integral to calculate the degree of irreversible thermal damage of inflamed and normal tissue. The Arrhenius damage integral is an empirical relationship that determines the degree of thermal damage as a function of temperature in biological tissue. Finally, the thermally damaged region ratio of inflamed and normal tissue was quantitatively analyzed using the Arrhenius thermal damage ratio proposed by Paik et al. [21]. The Arrhenius thermal damage ratio is a variable that quantitatively calculates the ratio of the volume of the thermally damaged region to the volume of total inflamed tissue.

## 2. Materials and Methods

### 2.1. Laser-Induced Heat Transfer Analysis

In this study, the Pennes bioheat equation was used to calculate the temperature distribution of tissue [22]. In addition, to apply the heat source by the laser, the final equation was expressed as Equation (1) [21].
(1)$$\rho c_{p}\frac{\partial T}{\partial t}=k\nabla^{2}T+q_{b}+q_{m}+q_{l}$$
(2)$$q_{b}=\rho_{b}\omega_{b}c_{p,b}\left(T_{b}-T\right)$$
(3)$$q_{l}=\left(1-R_{t}\right)\cdot \mu_{a}\frac{P_{l}\cdot cos\theta }{\pi r_{l}^{2}}e^{-\mu_{tot}\left(-\left(x+dx\right)sin\theta -\left(z-dz\right)cos\theta \right)}\cdot e^{-\frac{{\left(\left(x+dx\right)cos\theta -\left(z-dz\right)sin\theta \right)}^{2}+y^{2}}{r_{l}^{2}}}$$
where *ρ*, *c_p_*, and *k* denote density, specific heat, and thermal conductivity, respectively. *q_b_* represents the heat transfer term due to blood flow and is expressed as in Equation (2), where *ω_b_* and *T_b_* are blood perfusion rate and blood temperature, respectively. *q_m_* represents the metabolic heat source, and *q_l_* is the heat source caused by the laser and is expressed as Equation (3). *q_l_* considers the reflectivity when laser contacts the surface, the amount of energy according to the irradiation area, the energy attenuation according to the radial and depth directions, and the effect of the irradiation angle [23,24]. In this equation, *μ_a_*, *μ_tot_*, *R_t_*, *θ*, *P_l_*, and *r_l_* represent the absorption coefficient, attenuation coefficient, reflectivity, laser irradiation angle, laser power, and laser radius, respectively. Furthermore, *dx* and *dz* are differential lengths to adjust the laser irradiation position.
(4)$$R_{t}=R_{1}+R_{2},\;R_{1}={\left[\frac{\sqrt{n_{2}^{2}-{\left(n_{1}sin\theta \right)}^{2}}-n_{1}cos\theta }{\sqrt{n_{2}^{2}-{\left(n_{1}sin\theta \right)}^{2}}+n_{1}cos\theta }\right]}^{2}$$

Total reflectance *R_t_* is calculated as the sum of specular *R*_1_ and diffuse *R*_2_, as shown in Equation (4). *R*_1_ is calculated through the laser irradiation angle *θ* and the refractive indexes of air (*n*_1_) and inflamed tissue (*n*_2_), where *n*_1_ and *n*_2_ are 1 and 1.373, respectively [25,26]. *R*_2_ shows different values depending on the laser wavelength and has a value of 0.28 at 630 nm to be used in this study [27].

### 2.2. Inflamed Tissue Elimination Analysis

To determine the degree of thermally damaged tissue region based on the calculated temperature distribution, the Arrhenius damage integral was applied, as shown in Equation (5) [28]. This equation is calculated as a function of temperature and exposure time and is used to determine the degree of irreversible thermal damage to biological tissue.
(5)$$\Omega \left(t\right)=\int_{0}^{t}Ae^{-\frac{E_{a}}{RT\left(t\right)}}\;dt$$
where Ω, *A*, *E_a_*, *R*, and *T*(*t*) denote the degree of irreversible thermal damage to biological tissue, frequency factor, activation energy, ideal gas constant (8.314 J/mol∙K), and temperature at time *t*, respectively. In this study, *A* was applied as 2.84 × 10^99^ s^−1^ and *E_a_* as 0.618 MJ/mol [29]. If the calculated Ω is greater than or equal to 1, irreversible damage has occurred.

After calculating Ω at all points in the tissue, the Arrhenius variable proposed by Paik et al. [21] was used to quantify the thermally damaged region. The Arrhenius thermal damage ratio (*ϕ_Arrh_*), which determines the percentage of necrotic volume within the inflamed tissue, can be calculated as the ratio of the volume with Ω > 1 to the total inflamed volume, as shown in Equation (6). In addition, the normal tissue Arrhenius thermal damage ratio (*ϕ^N^_Arrh_*), which determines the percentage of the necrotic volume of normal tissue, can be calculated as the ratio of the volume of normal tissue with Ω > 1 to the volume of normal tissue with Ω > 1, as shown in Equation (7). Here, the area of normal tissue surrounding the inflamed tissue was limited to the gingival adjacent to the inflamed tissue, and its width was set to 50% of the width of the inflamed tissue [30].
(6)$$\phi_{Arrh}=\frac{\mathrm{inflammation}\;\mathrm{volume}\;\mathrm{at}\;\Omega >1}{\mathrm{total}\;\mathrm{inflammation}\;\mathrm{volume}}$$
(7)$$\phi_{Arrh}^{N}=\frac{\mathrm{normal}\;\mathrm{tissue}\;\mathrm{volume}\;\mathrm{at}\;\Omega >1}{\mathrm{Total}\;\mathrm{normal}\;\mathrm{tissue}\;\mathrm{volume}}$$

### 2.3. Numerical Conditions

This study analyzed the degree of photothermal therapy for peri-implantitis through numerical analysis. Figure 1 shows a schematic of the numerical analysis. Numerical modeling was implemented in three dimensions, and it was assumed that peri-implantitis occurred in a cone shape between the right side of the implant surface and the gums. The implant consists of a crown, abutment, and artificial tooth, with vertical lengths of 8.5 mm, 2 mm, and 13 mm, respectively. The upper part of the crown was set as an air zone, and the peri-implantitis was set to 1.5 mm in length and 9 mm in depth. For the gingival, it was set to have a thickness of 1 mm, and the alveolar bone is located at the bottom. The area of the entire tissue was set as a cube with a width, length, and height of 30 mm. The physical properties of all components are summarized in Table 1.

The numerical analysis was performed by varying the laser irradiation angle, irradiation position, and power. For the laser irradiation angle, it was set at 5° intervals from 15° to 40°. The minimum laser irradiation angle in this study was set at 15°, as the geometric angle of the implant crown area is close to 15°. Also, since the angle of the inflamed tissue relative to the vertical is close to 40°, the maximum laser irradiation angle was set to 40°. The irradiation position was set at 0.3 mm intervals from 0.1 mm to 1.3 mm relative to the outer edge of the inflamed tissue. For the laser power, it was set from 0 W to 4 W in 0.04 W increments for a total of 101 cases. For the irradiating laser, a 630 nm continuous wave laser with a Gaussian distribution was used, with a radius of 0.2 mm and an irradiation time of 300 s. All numerical simulation conditions are summarized in Table 2.

### 2.4. Numerical Model Validation

The numerical analysis was performed using the commercial program COMSOL Multiphysics 6.1. For the validation of the numerical modeling proposed in this study, the error depending on the mesh number and iteration was calculated. For the mesh number, the results converge at around 2.2 million and above. By checking the error as a function of the iteration number, it was found that the error converges to 10^−4^% or less from 34 iterations onwards. This confirms the validity of the numerical modeling proposed in this study.

## 3. Results and Discussion

### 3.1. Confirmation of Inflamed Tissue Elimination for Various Laser Irradiation Positions and Powers

Before confirming the thermally damaged region ratio of inflamed tissue, the temperature distribution and the thermally damaged range within the tissue according to various *d_l_* and *P_l_* were confirmed. Figure 2 shows the temperature distribution and thermally damaged range of the tissue for different *d_l_* and *P_l_* after fixing *θ* at 30°. In each graph, the *x*-axis represents the *x*-direction, and the *y*-axis represents the *z*-direction. Each component is represented by a white line; the black line in the figure indicates that Ω is 1. First, the temperature does not rise to the lowest point due to the geometry of the inflamed tissue. Generally, it was found that the thermally damaged range of inflamed tissue decreased as *d_l_* increased. This is because as the *d_l_* increases, the applied location of the laser moves closer to the implant, increasing the amount of laser heat absorbed by the implant. This means that the light dose was insufficient to raise the temperature of the inflamed tissue. This phenomenon can be seen in more detail by looking at the maximum depth of the line where Ω is 1. Also, as expected, an increase in *P_l_* increases the thermally damaged range of inflamed tissue. However, as the range of inflamed tissue increases, the range of surrounding normal tissue also increases. In order to raise the temperature of the deepest part of the inflamed tissue, the laser intensity must be excessively high, but the temperature rise in the surrounding normal tissue is also excessive, and thermal damage is inevitable. So, increasing *P_l_* unconditionally will not always maximize the therapeutic effect, which will be discussed in more detail later.

### 3.2. Confirmation of Inflamed Tissue Elimination for Various Laser Irradiation Angles and Powers

After fixing the *d_l_*, the effect of *θ* was also investigated. Figure 3 shows the temperature distribution and thermally damaged range of the tissue for different *θ* and *P_l_* after fixing dl at 0.7 mm. As mentioned above, due to inflammation and structural issues with the implant, the lowest laser irradiation angle was set at 15°. The smaller *θ* indicates that the laser is irradiated closer to the vertical. As can be seen in the figure, the thermally damaged range of inflamed tissue increases as *θ* decreases. This is because even if the same *P_l_* is applied, the amount of energy applied per unit area varies depending on *θ*, and the energy penetration length in the depth direction also varies. If *θ* is excessively increased, the penetration of the laser energy will be directed to the implant rather than the inflamed tissue, reducing the amount of energy absorbed by the inflamed tissue. Therefore, selecting treatment conditions to decrease *θ* is considered favorable in terms of increasing the temperature inside the inflamed tissue.

### 3.3. Elimination Ratio for Inflamed Tissue and Normal Tissue

Based on the calculated temperature distribution of inflamed and normal tissue, the degree of thermally damaged region was calculated using the Arrhenius thermal damage integral, and the thermally damaged region ratio of inflamed and normal tissue was identified using *ϕ_Arrh_* and *ϕ^N^_Arrh_*. Figure 4 shows *ϕ_Arrh_* as a function of *P_l_* and *d_l_* for each *θ*. In the graph, the *x*-axis represents laser power, and the *y*-axis represents *ϕ_Arrh_*. As shown in Figure 4, in all the cases, the thermally damaged region in inflamed tissue was less than 10% when the *d_l_* was the largest, i.e., the point closest to the implant (*d_l_* = 1.3 mm). This is because, as mentioned in Section 3.1, the laser energy is not absorbed by the inflamed tissue and is directed toward the implant. For the same *P_l_*, for *θ* ≤ 20°, the thermally damaged region ratio of inflamed tissue was maximized at a *d_l_* of 0.7 mm, whereas for *θ* > 25°, it was maximized at a *d_l_* of 0.4 mm. From Figure 4, the overall trend shows that as the laser irradiation location is moved towards the implant, the irradiation angle *θ* must be reduced (the closer the laser to the vertical) to increase the percentage of inflamed tissue eliminated. When *θ* is relatively small, the laser energy is applied in a relatively vertical direction, so it should be directed at the top center of the inflamed tissue to ensure that as much of the inflamed tissue as possible absorbs the laser energy. On the other hand, as *θ* increases, the penetration direction of the laser energy changes, so it is necessary to irradiate the laser from the relative outside of the inflamed tissue in order for a large portion of the inflamed tissue to absorb the laser energy. It can also be seen that when the laser irradiation location is between the center and the outside of the inflamed tissue, i.e., 0.1 mm < *d_l_* < 0.7 mm, the irradiation angle does not make a significant difference. Furthermore, it was found that *ϕ_Arrh_* increases at the same *P_l_* as *θ* decreases, which is due to the fact that the applied laser energy per unit area increases as *θ* decreases.

When performing treatment, it is important to consider not only the inflamed tissue but also the temperature of the surrounding gums and alveolar bone. Even if the laser energy is applied only to the inflamed tissue, if there is an excessive temperature increase within the inflamed tissue, the normal tissue may be damaged by heat transfer, so it is necessary to quantitatively confirm this.

Figure 5 shows *ϕ^N^_Arrh_* as a function of *P_l_* and *d_l_* for each *θ*. In the graph, the *x*-axis represents laser power, and the *y*-axis represents *ϕ^N^_Arrh_*. As shown in the figure, the highest thermally damaged region ratio of normal tissue was observed at the location closest to normal tissue, i.e., the smaller *d_l_*. The thermally damaged region ratio of normal tissues tended to decrease as the *d_l_* increased because the laser irradiation location is farther away from the normal tissue as the *d_l_* increases, and the laser energy cannot affect it. Therefore, *ϕ_Arrh_* and *ϕ^N^_Arrh_* should be considered simultaneously to identify the conditions that maximize the thermally damaged region of inflamed tissue while minimizing that of normal tissue.

As treatment should be performed while minimizing *ϕ^N^_Arrh_*, both *ϕ_Arrh_* and *ϕ^N^_Arrh_* need to be considered simultaneously. Comparing the damage to normal tissue with the damage to inflamed tissue, it appears optimal to irradiate at *d_l_* = 1.0 mm. For example, for *θ* = 15°, the ratio of *ϕ_Arrh_* to *ϕ^N^_Arrh_* calculated based on *P_l_* = 4 W is 1.23 for a *d_l_* of 0.1 mm and 1.88 for a *d_l_* of 1.0 mm. By calculating this trend for all cases, it was found that the ratio of *ϕ_Arrh_* to *ϕ^N^_Arrh_* is maximized when *d_l_* is 1.0 mm in all cases. Based on these results, it seems to be more beneficial from a therapeutic perspective to move the laser irradiation location closer to the implant.

## 4. Conclusions

In this study, the thermally damaged region ratio of inflamed and normal tissue under various treatment conditions was analyzed theoretically and numerically for peri-implantitis removal using photothermal therapy. Treatments were performed by varying the location, angle, and power of the laser irradiation, and the temperature distribution in the tissue was calculated for each condition.

The results of this study within the numerical analysis conditions presented indicate that considering only the thermal damage of inflamed tissue, the maximum thermally damaged region occurs at *d_l_* = 0.7 mm when *θ* is 20° or less and at *d_l_* = 0.4 mm when *θ* is 25° or more. However, when the degree of thermal damage of normal tissue is considered simultaneously, it was found that the treatment effect is maximized when *d_l_* is 1.0 mm for all *θ*. Furthermore, the results showed that laser irradiation at the point closest to the implant and at the point closest to the gingival had a very low therapeutic effect. The results presented in this study are expected to be helpful in establishing more accurate and strict treatment conditions for future laser treatment of peri-implantitis.

## Figures and Tables

**Figure 1 biomedicines-12-01976-f001:**
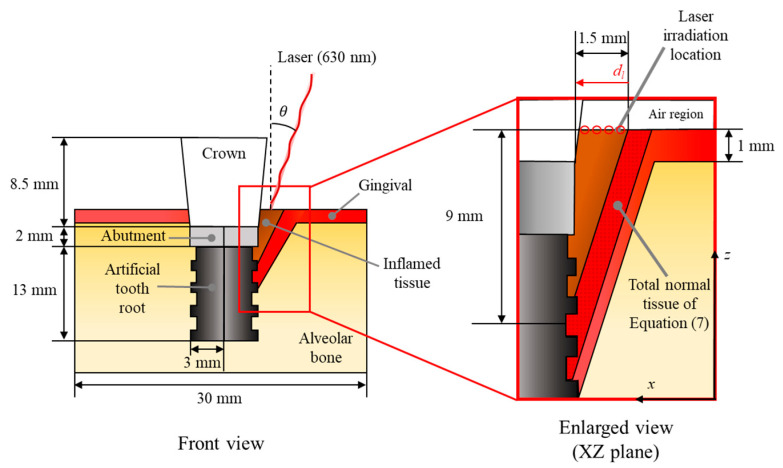
Schematic of numerical model.

**Figure 2 biomedicines-12-01976-f002:**
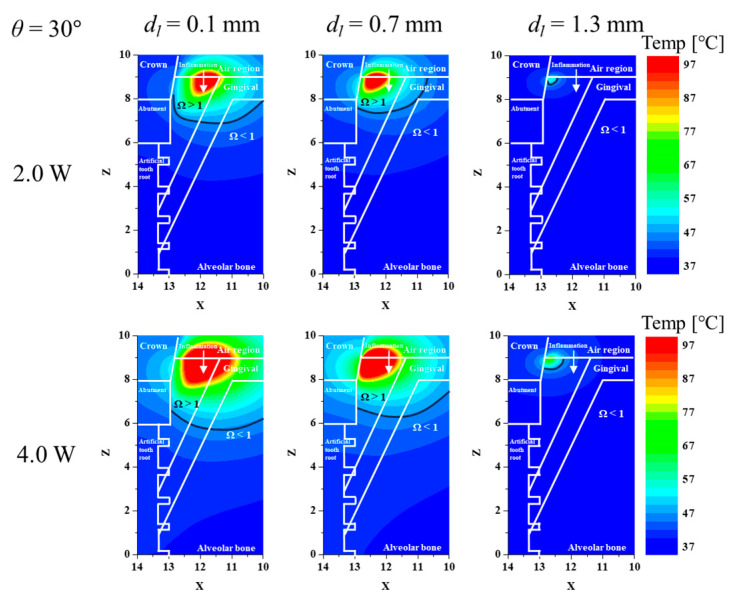
Temperature and Ω for various laser irradiation locations and powers (*θ* = 30°, XZ plane (*y* = 0)).

**Figure 3 biomedicines-12-01976-f003:**
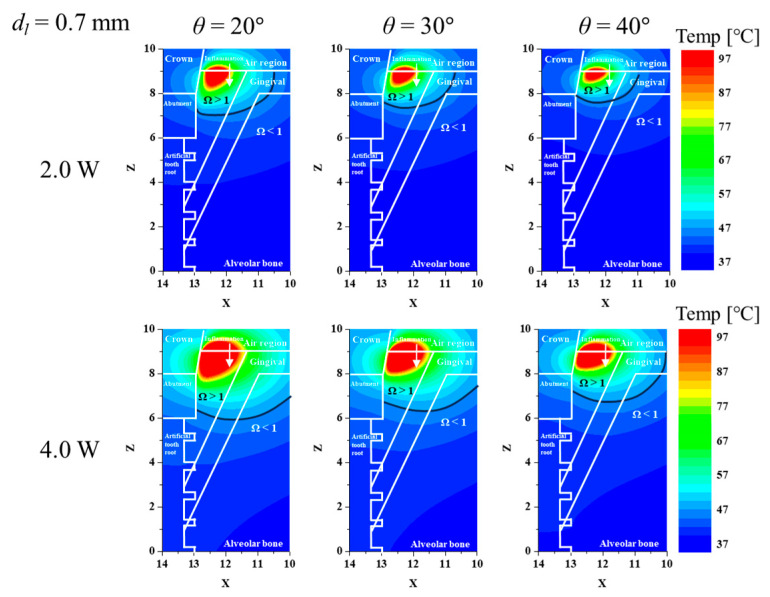
Temperature and Ω for various laser irradiation angles and powers (*d_l_* = 0.7 mm, XZ plane (*y* = 0)).

**Figure 4 biomedicines-12-01976-f004:**
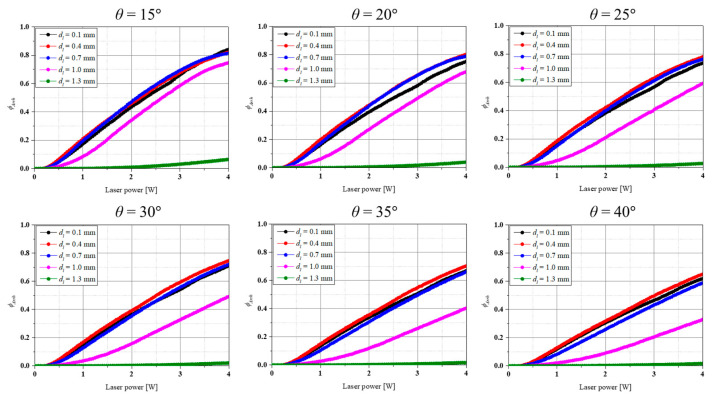
Arrhenius thermal damage ratio for various irradiation locations at each irradiation angle.

**Figure 5 biomedicines-12-01976-f005:**
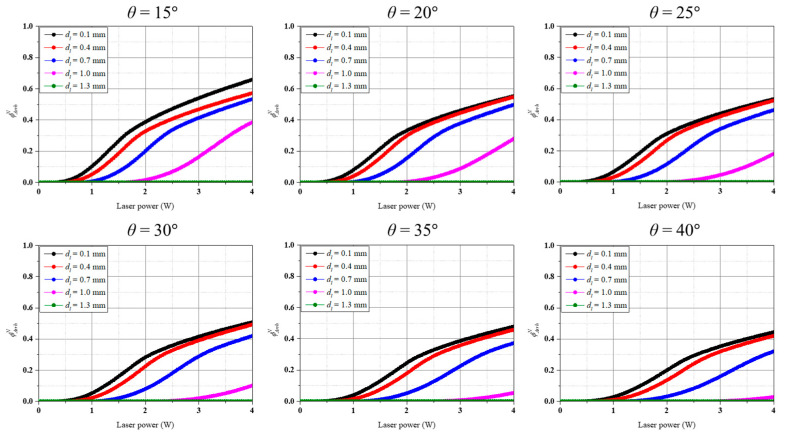
Normal tissue Arrhenius thermal damage ratio for various irradiation locations at each irradiation angle.

**Table 1 biomedicines-12-01976-t001:** Various properties of each implant component and tissues [28,31,32,33,34,35,36,37].

	*ρ* (kg/m^3^)	*c_p_* (J/kgK)	*k* (W/mK)	*μ_a_* (1/cm)	*μ*′_s_ (1/cm)	*ω_b_* (1/s)	*q_m_* (W/m^3^)
Inflamed tissue	1080	3500	0.48	2.16	17.03	0.009	65,400
Gingival	1000	4200	0.63	0.53	3.817	0.0076	1091
Alveolar Bone	2060	1260	0.38	0.596	22.97	0.00369	-
Crown (Zirconia)	6080	450	2.80	0.10	20.43	-	-
Abutment (Zirconia)	6080	450	2.80	0.10	20.43	-	-
Artificial tooth root (Ti-6Al-4V)	4420	546	7.00	789,500	≈0	-	-
Air	1.205	1.006	0.0256	0	0	-	-

**Table 2 biomedicines-12-01976-t002:** Parameters of numerical simulation.

Parameter	Case	Number	Remarks
Laser irradiation angle (*θ*)	15 to 40°	6	Interval: 5°
Laser irradiation location (*d_l_*)	0.1 to 1.3 mm	5	Interval: 0.3 mm
Laser Power (*P_l_*)	0.0 to 4.0 W	101	Interval: 0.04 W

## Data Availability

The original contributions presented in the study are included in the article, further inquiries can be directed to the corresponding author/s.

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
