# Peer review of "Analysis of Peri-Implantitis Photothermal Therapy Effect According to Laser Irradiation Location and Angle: A Numerical Approach"

_biomedicines, 2024, doi:10.3390/biomedicines12091976_

Round 1
Reviewer 1 Report
Comments and Suggestions for Authors
The theoretical model for the practical implication of photothermal therapy (PTT) in dental therapy/inflammation is promising. The discussion part can be written based on its prospect and usability. The selective comments for revision are the following:
1) In the Introduction part, the Arrhenius thermal theory and thermal damage value can be included briefly for the reader.
2) Figure 1 and other figures' descriptions should be detailed to understand the model.
3) The conclusion part needs to be concise. Some of the repetitions from the abstract can be deleted.
Author Response
Response to Reviewer 1 Comments
The theoretical model for the practical implication of photothermal therapy (PTT) in dental therapy/inflammation is promising. The discussion part can be written based on its prospect and usability. The selective comments for revision are the following:
1) In the Introduction part, the Arrhenius thermal theory and thermal damage value can be included briefly for the reader.
Response #1: Added a brief explanation of Arrhenius thermal theory and thermal damage values to the Introduction part.
|
Related Parts |
|
Page 2 Line 59: The results were then applied to Arrhenius damage integral to calculate degree of irreversible thermal damage of inflamed and normal tissue. Arrhenius damage integral is an empirical relationship that determines the degree of thermal damage as a function of temperature in biological tissue. Finally, thermally damaged region ratio of inflamed and normal tissue was quantitatively analyzed using the Arrhenius thermal damage ratio proposed by Paik et al. [19]. The Arrhenius thermal damage ratio is a variable that quantitatively calculates the ratio of the volume of thermally damaged region to the volume of total inflamed tissue. |
2) Figure 1 and other figures' descriptions should be detailed to understand the model.
Response #2: Added explanations for Figure 1 and other figures.
|
Related Parts |
|
Page 3 Line 111: This study analyzed the degree of photothermal therapy for peri-implantitis through numerical analysis. Figure 1 shows a schematic of the numerical analysis. Numerical modeling was implemented in three dimensions, and it was assumed that peri-implantitis occurred in a cone shape between the right side of implant surface and the gums. The implant consists of a crown, abutment, and artificial tooth, with vertical lengths of 8.5 mm, 2 mm, and 13 mm, respectively. The upper part of the crown was set as an air zone, and the peri-implantitis was set to 1.5 mm in length and 9 mm in depth. For the gingival, it was set to have a thickness of 1 mm, and the alveolar bone is located at the bottom. The area of the entire tissue was set as a cube with a width, length, and height of 30 mm. The physical properties of all components are summarized in Table 1. |
|
Page 4 Line 146: In each graph, the x-axis represents the x-direction length, and the y-axis represents the z-direction length. Each component is represented by a white line, |
|
Page 6 Line 181: In the graph, the x-axis represents laser power, and the y-axis represents ϕArrh. |
|
Page 7 Line 209: In the graph, the x-axis represents laser power, and the y-axis represents ϕNArrh. |
3) The conclusion part needs to be concise. Some of the repetitions from the abstract can be deleted.
Response #3: We've simplified the conclusion more as you suggested.
|
Related Parts |
|
Page 8 Line 231: In this study, thermally damaged region ratio of inflamed and normal tissue under various treatment conditions were analyzed theoretically and numerically for peri-implantitis removal using photothermal therapy. Treatments were performed by varying the location, angle, and power of the laser irradiation, and the temperature distribution in the tissue was calculated for each condition. The results of this study within the numerical analysis conditions presented indicate that, considering only the thermal damage of inflamed tissue, maximum thermally damaged region occurs at dl = 0.7 mm when θ is 20° or less and at dl = 0.4 mm when θ is 25° or more. However, when the degree of thermal damage of normal tissue is considered simultaneously, it was found that the treatment effect is maximized when dl is 1.0 mm for all θ. Furthermore, the results showed that laser irradiation at the point closest to the implant and at the point closest to the gingival had a very low therapeutic effect. The results presented in this study are expected to be helpful in establishing more accurate and strict treatment conditions for future laser treatment of peri-implantitis. |
Reviewer 2 Report
Comments and Suggestions for Authors
This report is apparently intended to describe the potential use of irradiation involving lasers to alleviate what is termed ‘inflammation’ in dental implants. It is never made entirely clear what causes this inflammation: infection, an allergic response, etc. The Abstract uses undefined terms: ‘Arrhenius damage integral’ and ‘Arrhenius thermal damage ratio’. Ths Abstract should be understood by all readers. What is ‘the death rate of inflammation’? Inflammation is a property of tissues. Tissues may die but what does ‘death of inflammation’ mean? This confusion occurs frequently throughout the report.
In the text, there is often no spacing between a word and the ‘[‘ symbol, e.g., on lines 34 and 37. A space should be inserted in such places. If ‘physical removal’ of relevant sites is difficult because such sites are ‘difficult to see’ (line 35), how will the operator know where to direct the laser? What is ‘the temperature distribution of inflammation (lines 55-56)? Does this mean the temperature of inflamed tissues? What is ‘the death rate of inflammation’ (line 59)? Tissues can die, but inflammation is a property of tissues. Does this mean that inflammation is reduced or that inflamed tissues can die? The Arrenhius damage integral needs to be defined.
What does ‘apply the heat source by the laser’ mean (line 65)? Does this mean an evaluation of elevated temperatures induced by irradiation? The term Ω is said to indicate the ‘damage degree of inflammation’ (line 85). What does this mean? What is ‘the damage degree’?
How is inflammation damaged? Material in lines 89-98 appears to be important to the argument but it is never properly explained. The term ‘the death rate of inflammation’ also occurs in lines 134 and 138. The term ‘Death’ is usually used to describe loss of the ability of cells, tissues or organisms to proliferate. Does this mean a decrease in inflammation?
This issue occurs elsewhere. To what does ‘death range’ refer in lines 135 and 155? Does line 167 mean the temperature distribution in inflamed tissue and normal tissue? Line 174: ‘death rate’ of what? Line 182 discusses irradiation ‘from the relatively outside of the tumor’. What tumor? The term ‘tumor’ is also used in line 183.
It might be better if the authors discuss ‘sites of inflammation’ in this report. There needs to be a distinction between ‘death of inflamed tissues and ‘a decrease in inflammation’. The authors do not always make it clear to which of these they are referring. For example, line 226 indicates ‘death of inflammation’ without telling readers whether this means decreased inflammation or death of inflamed tissues.
Comments on the Quality of English Language
Generally adequate
Author Response
Response to Reviewer 2 Comments
This report is apparently intended to describe the potential use of irradiation involving lasers to alleviate what is termed ‘inflammation’ in dental implants. It is never made entirely clear what causes this inflammation: infection, an allergic response, etc. The Abstract uses undefined terms: ‘Arrhenius damage integral’ and ‘Arrhenius thermal damage ratio’. Ths Abstract should be understood by all readers. What is ‘the death rate of inflammation’? Inflammation is a property of tissues. Tissues may die but what does ‘death of inflammation’ mean? This confusion occurs frequently throughout the report.
In the text, there is often no spacing between a word and the ‘[‘ symbol, e.g., on lines 34 and 37. A space should be inserted in such places. If ‘physical removal’ of relevant sites is difficult because such sites are ‘difficult to see’ (line 35), how will the operator know where to direct the laser? What is ‘the temperature distribution of inflammation (lines 55-56)? Does this mean the temperature of inflamed tissues? What is ‘the death rate of inflammation’ (line 59)? Tissues can die, but inflammation is a property of tissues. Does this mean that inflammation is reduced or that inflamed tissues can die? The Arrenhius damage integral needs to be defined.
Response #1:
1) First, a space was inserted between every word and ‘[‘.
2) In the case of physical removal, as mentioned in the manuscript, it is difficult to treat “difficult to see.” In the case of laser treatment, it is difficult to directly irradiate the laser to a point that is difficult to see in the same way. However, internally located inflammation is easier to treat with a laser because, unlike physical removal, the laser energy penetrates into the tissue, even if the laser is applied to the surface.
3) "the temperature distribution of inflammation" means the temperature distribution inside the inflamed tissue, and the temperature distribution inside the tissue was calculated by theoretical calculations. As you said, it is correct to mean inflamed tissue.
4) For "the death rate of inflammation", it refers to the rate of irreversible thermal damage to inflamed tissue. We apologize for the confusion. We've changed inflammation to inflamed tissue everywhere.
5) The Arrhenius damage integral is an equation mainly used in the field of biological heat transfer. This equation is an empirical relationship that determines the degree of thermal damage as a function of temperature in biological tissue.
We added a brief description of Arrhenius damage integral.
|
Related Parts |
|
Page 2 Line 60: Arrhenius damage integral is an empirical relationship that determines the degree of thermal damage as a function of temperature in biological tissue. |
|
Page 3 Line 90: This equation is calculated as a function of temperature and exposure time and is used to determine the degree of irreversible thermal damage of biological tissue. |
What does ‘apply the heat source by the laser’ mean (line 65)? Does this mean an evaluation of elevated temperatures induced by irradiation? The term Ω is said to indicate the ‘damage degree of inflammation’ (line 85). What does this mean? What is ‘the damage degree’?
Response #2:
1) 'apply the heat source by the laser' means that the heat transfer term due to the laser is added to the various heat transfer terms. In terms of formula, it is expressed as above because the heat transfer term by the laser must be included.
2) Ω represents the degree of irreversible thermal damage to biological tissue. We apologize for the confusion in the wording. The phrase has been changed as follows.
|
Related Parts |
|
Page 3 Line 92: Where Ω, A, Ea, R, and T(t) denote the degree of irreversible thermal damage to biological tissue, frequency factor, ~ |
How is inflammation damaged? Material in lines 89-98 appears to be important to the argument but it is never properly explained. The term ‘the death rate of inflammation’ also occurs in lines 134 and 138. The term ‘Death’ is usually used to describe loss of the ability of cells, tissues or organisms to proliferate. Does this mean a decrease in inflammation?
Response #3: It seems that my limited understanding of the biological aspects led me to phrase it incorrectly. I apologize for that. The final value calculated in Arrhenius thermal damage is Ω, which signifies irreversible thermal damage. In this study, we refer to irreversible damage as death. We apologize again for any confusion caused by this terminology. We have now replaced the term "death" with "thermally damaged" throughout the manuscript.
This issue occurs elsewhere. To what does ‘death range’ refer in lines 135 and 155? Does line 167 mean the temperature distribution in inflamed tissue and normal tissue? Line 174: ‘death rate’ of what? Line 182 discusses irradiation ‘from the relatively outside of the tumor’. What tumor? The term ‘tumor’ is also used in line 183.
Response #4:
1) ‘Death range’ refers to the irreversible thermal damaged range. Sorry for the confusion, as we mentioned in response #3. ‘Death rate’ also refers to the thermally damaged region ratio.
2) As you said, line 167 means temperature distribution in inflamed tissue and normal tissue.
|
Related Parts |
|
Page 6 Line 178: Based on the calculated temperature distribution of inflamed and normal tissue, the degree of thermally damaged region was calculated~ |
3) In line 174, 'death rate' means thermally damaged region ratio of inflamed tissue. We have fixed the relevant part.
|
Related Parts |
|
Page 6 Line 186: For the same Pl, for θ ≤ 20°, the thermally damaged region ratio of inflamed tissue was maximized at a dl of 0.7 mm, whereas for θ > 25°, it was maximized at a dl of 0.4 mm. |
4) The part marked as ‘tumor’ is a typo. It means inflamed tissue. We have corrected the typos.
|
Related Parts |
|
Page 6 Line 194: ~ irradiate the laser from the relatively outside of the inflamed tissue in order for a large portion of the inflamed tissue to absorb the laser energy. |
It might be better if the authors discuss ‘sites of inflammation’ in this report. There needs to be a distinction between ‘death of inflamed tissues and ‘a decrease in inflammation’. The authors do not always make it clear to which of these they are referring. For example, line 226 indicates ‘death of inflammation’ without telling readers whether this means decreased inflammation or death of inflamed tissues.
Response #5:
First, the area where inflammation occurred was specified.
|
Related Parts |
|
Page 4 Line 113: ~ and it was assumed that peri-implantitis occurred in a cone shape between the right side of implant surface and the gums. |
As you mentioned, there was some confusion about the death of inflamed tissue and the decrease in inflammation. Since the calculations in this study were done in relation to thermally induced damage, we have changed the word death to refer to thermal damage as mentioned in the above response.
Round 2
Reviewer 2 Report
Comments and Suggestions for Authors
While this report is substantially improved, it would be helpful to have it edited by someone with a bit more experience with English usage. There remain passages that are difficult for the reader to understand. An example: lines 76-80: ‘The reflectivity [considered] when the laser contacts the surface, the amount of energy depending on the irradiated area, the energy attenuation depending on the radial and depth directions, and the influence of the irradiation angle are each [considered] [21, 22]. [Where] μa, μtot, Rt, θ, Pl, and rl represent the absorption coefficient, attenuation coefficient, reflectivity, laser irradiation angle, laser power, and laser radius, respectively.’ This material is among the sections that are difficult for the reader to decipher. I suspect that the first [considered] should be deleted, the second use of ‘considered’ should perhaps be replaced with ’important’. The word ‘where’ can be replaced with ‘in this equation’. Another example occurs on line 143-145: ‘Before confirming thermally damaged region ratio of inflamed tissue, temperature distribution and thermally damaged range within the tissue according to various dl and Pl were confirmed.’ Perhaps the word ‘the’ should be inserted after the first use of ‘confirming’? The word ‘length’ can be removed from line 147. Line 152 ‘laser is insufficiently energized’ should be replaced with ‘the light dose was insufficient . . ’. There are other such examples where editing is needed. Another example (line 183): ‘the damaged ratio of inflamed tissue . . ‘ Ratio of what to what? A good editor will know what to do.
This is otherwise suitable for publication.
Comments on the Quality of English Language
See comments for authors
Author Response
Response to Reviewer 2 Comments
While this report is substantially improved, it would be helpful to have it edited by someone with a bit more experience with English usage. There remain passages that are difficult for the reader to understand. An example: lines 76-80: ‘The reflectivity [considered] when the laser contacts the surface, the amount of energy depending on the irradiated area, the energy attenuation depending on the radial and depth directions, and the influence of the irradiation angle are each [considered] [21, 22]. [Where] μa, μtot, Rt, θ, Pl, and rl represent the absorption coefficient, attenuation coefficient, reflectivity, laser irradiation angle, laser power, and laser radius, respectively.’ This material is among the sections that are difficult for the reader to decipher. I suspect that the first [considered] should be deleted, the second use of ‘considered’ should perhaps be replaced with ’important’. The word ‘where’ can be replaced with ‘in this equation’. Another example occurs on line 143-145: ‘Before confirming thermally damaged region ratio of inflamed tissue, temperature distribution and thermally damaged range within the tissue according to various dl and Pl were confirmed.’ Perhaps the word ‘the’ should be inserted after the first use of ‘confirming’? The word ‘length’ can be removed from line 147. Line 152 ‘laser is insufficiently energized’ should be replaced with ‘the light dose was insufficient . . ’. There are other such examples where editing is needed. Another example (line 183): ‘the damaged ratio of inflamed tissue . . ‘ Ratio of what to what? A good editor will know what to do.
This is otherwise suitable for publication.
Response #1: Thank you for your detailed review.
1) The explanation in lines 76-80 was intended to mean that each term is included in the equation. We changed that part as follows. Also, We changed “Where” to “In this equation” as you suggested.
|
Related Parts |
|
Page 3 Line 93: ql considers the reflectivity when laser contacts the surface, the amount of energy according to the irradiation area, the energy attenuation according to the radial and depth directions, and the effect of the irradiation angle [21,22]. |
|
Page 3 Line 95: In this equation, μa, μtot, Rt, θ, Pl, and rl represent the absorption coefficient ~ |
2) As you mentioned in lines 143-145, we inserted “the” after “confirming”. Also, we removed the word “length” in line 147.
|
Related Parts |
|
Page 5 Line 162: Before confirming the thermally damaged region ratio of inflamed tissue, temperature distribution and thermally damaged |
|
Page 5 Line 165 : In each graph, the x-axis represents the x-direction, and the y-axis represents the z-direction. |
3) As you mentioned in line 152, we changed “laser is insufficiently energized” to “the light dose was insufficient”. Additionally, in line 183, we changed “the damaged ratio of inflamed tissue” to “thermally damaged region in inflamed tissue”.
|
Related Parts |
|
Page 5 Line 171: This means that the light dose was insufficient to raise the temperature of the inflamed tissue. |
|
Page 7 Line 207: As shown in Figure 4, in all the cases, the thermally damaged region in inflamed tissue was ~ |